# Biological Activity of Berberine—A Summary Update

**DOI:** 10.3390/toxins12110713

**Published:** 2020-11-12

**Authors:** Anna Och, Rafał Podgórski, Renata Nowak

**Affiliations:** 1Chair and Department of Pharmaceutical Botany, Medical University of Lublin, 1 Chodźki St., 20-093 Lublin, Poland; renatanowak@umlub.pl; 2Department of Biochemistry, Institute of Medical Sciences, Medical College of Rzeszów University, 16c Rejtana St., 35-959 Rzeszów, Poland; rpodgorski@ur.edu.pl

**Keywords:** berberine, anticancer, metabolic syndrome, apoptosis, clinical trials, bioavailability

## Abstract

Berberine is a plant metabolite belonging to the group of isoquinoline alkaloids with strong biological and pharmacological activity. Currently, berberine is receiving considerable interest due to its anticancer activity based on many biochemical pathways, especially its proapoptotic and anti-inflammatory activity. Therefore, the growing number of papers on berberine demands summarizing the knowledge and research trends. The efficacy of berberine in breast and colon cancers seems to be the most promising aspect. Many papers focus on novel therapeutic strategies based on new formulations or search for new active derivatives. The activity of berberine is very important as regards sensitization and support of anticancer therapy in combination with well-known but in some cases inefficient therapeutics. Currently, the compound is being assessed in many important clinical trials and is one of the most promising and intensively examined natural agents.

## 1. Introduction

Alkaloids with strong activity have always been used in traditional medicine as plant extracts. Currently, with the new research methods, active compounds in extracts can be established and designed for new applications. One of them is berberine, which is currently receiving great interest due to its extremely promising biological and pharmacological activity (Figure 1). The earliest information on the medical use of *Rhizoma coptidis* containing berberine is dated in A.D. 200. [1]. Recent berberine research has not only confirmed the significance of its use in traditional Chinese medicine, where it was applied in various diseases, for example infections and gastrointestinal disorders [2], but has also proved its anticancer activity and effectiveness in neurological and metabolic disorders. Despite its poor bioavailability, limiting its application [3], berberine is currently being assessed in many important clinical trials, which encourages deeper examination of the mechanisms of its action and a search for new applications. The most promising is the anticancer use of berberine. Berberine not only possesses documented proapoptotic activity, which is in the focus of attention, but also seems to be a very important and promising compound in combined cancer treatment. Sensitization and elimination of drug resistance are very promising trends in the berberine research. What is more, berberine exhibits low toxicity towards healthy cells, which makes it safe for clinical use and proves its activity in biochemical disorders. Due to the low bioavailability and poor pharmacokinetic parameters, research of new forms of berberine administration and its new active derivatives has begun. The latest papers from 2019 and 2020 seem to focus on this priority and attempt to solve the problem to provide efficient berberine-based therapy.

## 2. Berberine in Cardiovascular and Metabolic Diseases

### 2.1. Cholesterol-Lowering Effect

Berberine has been clinically examined quite extensively as regards its beneficial influence on the cardiovascular system. Berberine has an antiarrhythmic effect, improves ejection fraction and enhances the function of the left ventricle and general physical capacity in congestive heart failure [4]. Berberine decreases blood pressure by reducing cholesterol via several mechanisms, e.g., it stimulates the capture of cholesterol in serum by liver, stimulates the disposal of LDL-C from blood [2,5], reduces the absorption of cholesterol in bowels, enhances cholesterol excretion in excrements and stimulates liver exchange of cholesterol and the formation of bile acid [6]. It also stimulates AMPK (protein kinase activated by 5′ adenosine monophosphate), which may limit fatty acid synthesis [7]. Reduced TC, TG and LDL-C concentration and an increased HDL-C concentration were noted after three months of application of the compound [8]. In HepG2 cells, i.e., a human hepatoma cell line in primary hepatocytes, berberine inhibited the synthesis of cholesterol and triglyceride. It also lowers cholesterol levels in vivo [2].

### 2.2. Antidiabetic Action

In type 2 diabetes mellitus, berberine was first reported in 1986 [9]. Indirect clinical investigations of the effect of berberine proved that it reduced alanine and aspartate transaminase levels in diabetic patients [9]. It was reported as a successful agent alleviating insulin resistance [10]. The hypoglycemic effect of berberine is comparable to metformin. Berberine, likewise to metformin, regulates a variety of effectors, such as AMPK and MAPK (mitogen activated protein kinase) [11]. Hyperinsulinemia and insulin resistance, typical for type 2 diabetes are crucial in polycystic ovary syndrome pathogenesis [12]. Hence, berberine is considered effective in polycystic ovary syndrome, but this issue needs further investigation [13]. It has been clinically confirmed that berberine enhances ovulation in polycystic ovary syndrome by reducing insulin resistance [14]. Moreover, the effect of berberine on the lipid profile in women with polycystic ovary syndrome is highly beneficial. The intake of 500 mg berberine for 3 months significantly improved the profile in treated patients. A higher pregnancy rate and a lower incidence of severe ovarian hyperstimulation syndrome were noticed as well [15]. In addition, the treatment with berberine instead of metformin entailed fewer adverse effects and reduced lipid parameters and BMI (Body Mass Index) [10].

### 2.3. Antiobesity Action

As mentioned previously, berberine is a potential drug for treatment of obesity. It acts by downregulation of adipogenesis and lipogenesis. This antiobesity activity is connected with the fact that berberine strongly decreased the size and number of droplets of lipid in the 3T3-L1 adipocyte cell line. The mitigation of high glucose-induced podocyte apoptosis after berberine treatment described currently is equally promising for diabetic patients. In this case, the alkaloid modulates autophagy and it proceeds via the mTOR/P70S6K/4EBP1 pathway [11]. A long-term body weight loss effect is observed after expousure to berberine. The effect is exerted through enhancing ATGL expression (AMPK-mediated) and increases the basal lipolysis state of triglycerides in adipocytes.

Crucial for the transcription factor of adipogenesis is PPARγ. Berberine inhibits adipocyte differentiation via PPARγ and C/EBPα pathways. Besides, berberine inhibits the proliferation and differentiation of preadipocytes [9]. With its regulatory function in insulin resistance and dyslipidemia, berberine can be a potential agent in metabolic syndrome. However, the overall effect of berberine in metabolic syndrome has not been systemically tested, partly because the preclinical models for metabolic syndrome are limited [16].

The anticancer activity of berberine via an effect on kinases, described in the further part of the article, is possibly also related to its effects on kinase-mediated lipid metabolic pathways. Insulin resistance development in obesity involves JNK kinases [17] and neurotransmitter excitotoxicity in ischemic conditions. [18]. Hong et al. described that berberine reduced the phosphorylation of JNK in gastric cancer cells [19], but the influence of berberine on kinases is strongly dependent on many conditions [20,21,22].

## 3. Berberine in Neurodegenerative and Neuropsychiatric Disorders

The ability to ameliorate hyperlipidemia and hyperglycemia make berberine supportive in neurologic disease [23,24]. Recent studies have shown that berberine exerts a protective effect on the central nervous system, which makes it a promising agent in disorders such as Alzheimer’s disease, cerebral ischemia, mental depression, anxiety and schizophrenia [25,26]. Berberine exerts a neuroprotective effect by regulating early immune activation of peripheral lymphocytes and immunotolerance in vivo [27]. However, this is not fully understood and there are reports on berberine exacerbating neurodegeneration [28].

Berberine also significantly decreases the production of kynurenine, which when increased, is metabolized to neurotoxic compounds (for example quinolinic acid), and influences glutamatergic neurotransmission [3,29]. It has been described that berberine inhibits the effects of reward after abuse of drugs such as cocaine, morphine and ethanol. It proceeds through downregulation of tyrosine hydroxylase expression or other mechanisms [23,30,31]. Researchers suggest that alkaloids may rapidly act like antidepressants; hence, it is indicated as a potential substance for the treatment of patients with major depression. Berberine, like other antidepressant drugs, affects sigma receptor 1. Studies also show that berberine can act as an antidepressant via the NF-κB (nuclear factor kappa-light-chain-enhancer of activated B cells) signaling pathway, which is activated by oxidative stress. This berberine antidepressant effect also results from its impact on the brain-derived neurotrophic factor—cAMP-response element—binding protein pathway. This well-known antidepressant pathway is crucial for the antidepressant action of drugs. Berberine acts by elevation of neurotrophic factor levels and restores the decreased level of its mRNA [3,29]. Berberine easily crosses the blood–brain barrier after systemic administration, which enhances its potential in treatment of neurological diseases [23,24].

## 4. Anticancer Activity of Berberine

The first study of the cytotoxic activity of berberine was published in 1986 [32]. Later studies demonstrated the cytotoxic activity of berberine towards many cancer cell lines such as the promyelocytic leukemia HL-60 cell line [33,34], uterine cancer HeLa cell line [35,36], lymphocytic leukemia L1210 cell line, myelomonocytic leukemia WEHI-3 cell line [37], myeloid leukemia K562 cell line [38], large intestine cancer HT29 cell line [39,40], bladder cancer BIU-87 and T24 cell lines [41], hepatoma HepG2 cell line, non-small cell lung cancer [20,42], Lewis lung cancer [43], astrocytoma G95/VGH and GBM 8401 cell lines [44], melanoma B16 cell line and model U937 cell line [45]. Berberine is cytotoxic towards cancer cell lines and this activity is dependent on the dose and time. Studies have shown many mechanisms of the anticancer activity of berberine. In the assessment of the levels of expression of a panel of 44 genes in the human colon adenocarcinoma HCA-7 cell line, berberine treatment resulted in downregulation of 33 genes differently involved in the cell cycle, differentiation and epithelial–mesenchymal transition in a time- and dose-dependent manner [46]. The therapeutic window of berberine in most cases is narrow and depends on the dosage and type of cells that are treated.

### 4.1. Cell Cycle Arrest

It has been shown that berberine in low concentrations arrests human cancer cells in the G1 phase, while high concentrations arrest the cell cycle in the G2/M phase [29,47].

Berberine has been shown to inhibit the cell cycle in the G1 phase by upregulation of the BTG2 gene (B cell translocation gene 2), which is a cell proliferation regulatory gene induced by the p53 protein. The arrest of the cell cycle in the G2/M phase is p53 independent [47,48,49].

The arrest of the cell cycle in the G0/G1 phase after exposure to berberine was reported in lymphocytic leukemia cell line L1210 [35] and bladder cancer cells BIU-87 and T24 cell lines [50]. Colon cancer cells exposed to berberine were characterized by G0/G1 phase cell cycle arrest with downregulation of the antiapoptotic gene BCL2 in a concentration-dependent manner [51,52,53].

Cyclins seem to be an important target for cell cycle arrest induced by berberine. Downregulation of cyclin D1 was observed after exposure to berberine in the G1 cell cycle phase [47]. Reduction of the expression of cyclin B1 by berberine and the increase in the expression of Wee1 can arrest tumor cells in the G1 and G2 phases [47,54]. G0/G1 arrest was observed in MDA-MB-231 and MCF-7 breast cancer cells after exposure to berberine, possibly due to a decrease in the level of the cell cycle regulatory protein cyclin B1. This effect was also dose dependent [47].

As reported by Chidambara et al., arrest of the cell cycle in the G2/M phase by berberine is dependent on the REV3 gene. Cells of the DT40 line deficient in REV3 are hypersensitive to berberine and their DNA undergoes double-strand breaks much more strongly than the DNA of wild-type cells after exposure to berberine [55]. Berberine-induced inhibition of the cell cycle in the G2/M phase has also been described in colorectal cancer cells of the SW480 line [56].

### 4.2. Apoptosis Induction

One of the most important and comprehensively examined processes triggered by exposure to berberine, as regards its anticancer activity, is apoptosis. Induction of a number of biochemical events i.e., a decrease in the mitochondrial membrane potential, release of cytochrome c, Bcl2 family proteins and caspase activation or PARP breakdown after exposure to berberine, confirms the proapoptotic abilities of berberine [57].

Berberine induces apoptosis in tumor cells, mainly via upregulation of proapoptotic genes and downregulation of antiapoptotic genes [34,58]. Changes in the gene expression in leukemic cells exposed to berberine showed that, despite low cytotoxicity of observed dosage, the compound significantly increased the expression of caspase genes CASP3, CASP8 and CASP9 and proapoptotic genes BAK1, BAX and BIK. Simultaneously, downregulation of the expression of antiapoptotic genes BCL2, BCL2L2, BNIP1 and BNIP3 was noticed. This indicates that gene regulation leads to the apoptosis caused by berberine [34].

In other studies, after exposure of cells from the HL-60 [33], U937 and B16 lines [45] to berberine, activation of protein caspase -3 and -9, an increase in Bax (Bcl2-associated X protein) and a decrease in the Bcl-2 protein level were reported [33]. Additionally, an increase in the level of important proapoptotic proteins taking part in apoptosis signaling pathways such as p53, Rb (retinoblastoma Protein), ATM (serine/threonine kinase), caspase-8, Fas Receptor (death receptor)/FasL (Fas ligand), BID (BH3 interacting-domain death agonist, a proapoptotic member of the Bcl-2 protein family) and TNF (tumor necrosis factor) was reported. On the other hand, a decrease in the level of c-IAP1 (inhibitor of apoptosis protein), XIAP (X-linked Inhibitor of apoptosis protein), Bcl-X and Survivin (antiapoptotic protein) was reported after exposure to berberine. Among other mechanisms, berberine was shown to regulate proapoptotic and antiapoptotic proteins through an increase in the level of ROS—an important agent in apoptosis regulation [22,58,59].

An important role in berberine-induced apoptosis is played by death receptors, known as TRAIL receptor 2. TRAIL (tumor necrosis factor related apoptosis inducing ligand) is an apoptosis-inducing ligand associated with the tumor necrosis factor. It is known that TRIAL has great potential in the treatment of cancer, as it induces apoptosis by binding to the aforementioned receptors that induce tumor cell death, i.e., the so-called “death receptors”—DR4 and DR5. TRAIL induces apoptosis selectively; however, the development of partial or complete resistance limits its use. Berberine shows synergy with TRAIL. Moreover, it sensitizes cancer cells with TRAIL resistance. In TRAIL-sensitive (MDA-MB-231) and -resistant (MDA-MB-468) human breast cancer cell lines, berberine synergized with TRAIL, but it also sensitized the resistant cells to TRAIL. The markers of the process were caspase-3 and PARP 9 Poly (ADP-ribose) polymerase 1 cleavage and p53. The berberine sensitization to TRAIL-induced apoptosis is not limited to TRAIL-resistant MDA-MB-468 breast cancer cells. Despite the moderate cytotoxic effect on breast cancer cell line 4T1 in vitro, berberine in combination with antiDR5 markedly inhibited primary growth of tumor and reduced tumor metastasis to the lungs [54,60].

### 4.3. Influence on MAPK

There are many scientific reports on the effects of berberine on mitogen-activated kinases (called MAP or MAPK), which are involved in directing cellular responses to a variety of stimuli. They regulate processes that are very important in carcinogenesis, e.g., gene expression, mitosis, apoptosis, proliferation and differentiation [29].

Berberine has been shown to modulate mitogen-activated protein kinase signaling pathways, such as extracellular signal regulated kinase 1/2 (ERK1/2), p38 MAPK (p38 mitogen-activated protein kinases) and c-Jun N-terminal kinase (JNK) pathways. Compounds modulating these pathways are noteworthy as potential anticancer drugs. The effect depends on the type of cell. Berberine activates MAPK in human colon cancer cells [61], non-small cell lung cancer cells and human hepatoma cells (HepG2) [20,21]. In turn, in human HeLa cervical carcinoma cells, berberine enhances phosphorylation of JNK and ERK1/2 but inhibits phosphorylation of p38 MAPK [62]. In a Hong et al.’s study, berberine reduced the p38 MAPK, ERK1/2 and JNK phosphorylation in gastric cancer cells [19]. The JNK/p38 MAPK signaling pathway is disrupted in many types of cancer [63]. It was found that berberine suppressed the invasion and migration of cancer cells through blocking the JNK/p38 signaling pathway in the gastric cancer SNU-1 cell line [22].

One of the more precisely described phenomena in this field of berberine activity is its influence on MAPK via the impact of micro RNA inhibiting the translation of certain proteins, the dysfunction of which plays a role in formation of, e.g., non-small cells lung cancer. Abnormal levels of these proteins are correlated with the tissue factor TF, which contributes to tumor metastasis of non-small cells lung cancer. It has been shown to activate signaling cascades, including MAPK. In human lung cancer A549 cells, apoptosis through the miR-19a/TF/MAPK signaling pathway has been described after exposure to berberine [21]. Berberine raises the level of miR-19a and lowers the level of TF, thus activating MAPK signaling leading to apoptosis of cancer cells [21].

The cyclin-dependent kinase inhibitor p21 (CIP1/WAF1) is involved in the cell cycle control, cell differentiation, apoptosis and DNA replication [27]. It is linked with p53 and FOXO3a in the control of cancer cell growth [64,65,66]. FOXO3a (human protein Forkhead box O3) is a transcription factor from a family of transcription factors with tumor suppressor activity. It is regulated by growth factor receptor-induced activation of the phosphatidylinositol 3-kinase (PI3-K)/Akt signaling pathway. Its activation is connected with apoptosis [67] and cell cycle arrest [68] and, in various types of cells, it is associated with tumor suppression. Inhibition of FOXO3a causes tumor progression [69]. Zheng et al. demonstrated that, in non-small cell lung cancer, berberine inhibited proliferation and induced apoptosis by activating the p38α MAPK signaling pathway, resulting in an increase in p53 and FOXO3a and induction of the cell cycle inhibitor p21 (CIP1/WAF1) [20].

### 4.4. Trancription Regulation

Berberine also exhibits activity against the very important transcription factor-1 (AP-1), which is closely related to neoplastic transformation. AP-1 consists of complexes comprising the following families of DNA-binding proteins: Fos family (c-Fos, Fra-1, FosB and Fra-2,), Jun family (c-Jun, JunD, JunB and v-Jun), ATF/cyclic AMP-responsive element-binding (b-ATF, ATF1–4, ATF-6 and ATFx) and Maf family (c-Maf, MafA, MafB, MafG/F/K and Nrl), which play a key role in inflammation, proliferation and apoptosis. The activity of AP-1 is regulated by, e.g., growth factors, infections, cytokines, UV radiation, or cellular stress [70].

Extrinsic carcinogens can induce increased AP1 activity [71]. Many human tumors overexpress members of the Jun protein family [72,73,74]. Overexpression of these proteins has been described in aggressive forms of lymphomas [75,76] and in breast cancer [74]. However, increased expression of c-Fos is observed in endometrial cancer and osteosarcoma, while decreased expression of c-Fos is associated with the progression of ovarian and gastric cancer [77]. The role of the Fos family in tumor development is thus tissue-specific [70]. Studies have shown that in general AP-1 activation depends on the type of extrinsic stimulus and the cellular condition may have different effects on the cell fate. It has been shown that the AP-1 protein was inhibited in hepatoma cells of the HepG2 line after exposure to berberine [39,78]. In turn, in oral administration of berberine, spontaneous metastasis of Lewis lung cancer cells from mediastinal lymph nodes to lung parenchyma was inhibited through AP-1 protein activation [43]. After oral administration of berberine, decreased expression of the C-fos proto-oncogene was described [54]. Thus, the influence of berberine on the AP1-protein family depends on the cell type and requires further investigations.

### 4.5. Inhibition of Metastasis

Berberine is a potential agent to halt or prevent metastasis. It acts at several points of cancer progression. One is its strong influence on matrix metalloproteinases—important proteins involved in degradation of the barrier of extracellular matrix—the first and important step in tumor cell metastasis.

Changed expression and levels of MMP activity are strongly involved in the development of many cancers. Increased MMP-2 activity is associated with poor prognosis in such types of cancer as colon, breast, melanoma, ovary, prostate and lung cancers [79]. Changes in MMP-2 activity may also be derived from changes in the levels of activation, inhibition and secretion or transcription of the MMP group of enzymes. MMP production in many cancers is elevated in the surrounding stromal tissue, but not in the tumor, and cases of metastasis are correlated with higher levels of MMP-2 mRNA in surrounding healthy tissue [80]. MMP-2 and MMP-9 can degrade type IV collagen, i.e., a major component of the basement membrane that is important for maintaining tissue organization and provides cell signaling and polarity. The degradation of the extracellular matrix allows cells to migrate from the tumor to form metastases. This is an essential step in the metastatic progression in most cancers [80]. Moreover, products of degradation of MMPs further promote MMP activity [81]. Berberine inhibits the expression of matrix metalloproteinase-2 (MMP-2) and matrix metalloproteinase-9 (MMP-9) in a time- and concentration-dependent manner. In tests on mice, berberine was found to reduce metalloproteinase levels in plasma [82].

The regulation of the expression of matrix metalloproteinases by berberine proceeds through the inhibition of the transfer of p-STAT3 (signal transducer and activator of transcription 3) to the nucleus. In colon cancer cells it significantly reduces the level of JAK2 (Janus kinase 2—A protein from the Janus kinase family) and STAT3 phosphorylation. Phosphorylated molecules p-JAK2 and p-STAT3 are significantly increased in colorectal cancer cells overexpressing COX2 (cyclooxygenase-2). Overexpression of COX2 induces the activation of JAK-STAT, which increases the levels of metalloproteinases—MMP-2 and MMP-9 in colon cancer cells. Berberine markedly decreased the levels of phosphorylated JAK2 and phosphorylated STAT3 in colorectal cancer cells and effectively interrupted the COX2/JAK/STAT signaling pathway [23,29], which was observed as a decrease in the level of metalloproteinases [54,83].

Berberine reduces protein levels of the STAT3 in nasopharyngeal carcinoma cells and blocks STAT3 activation induced by IL-6 secreted by tumor-associated fibroblasts. Similar to the Janus kinase family, the family of kinases of the transcription factors of the STAT plays an important role in the immunity process, cell division, cell death and tumor formation [84].

### 4.6. Inhibition of Angiogenesis

Another mechanism of berberine in the antiprogression process is inhibition of angiogenesis.

Additionally, in this case, the influence on metalloproteinases 2 and 9 plays a crucial role. MMP-2 plays an important role in the formation of new blood vessels in tumors by supporting the migration of endothelial cells. It is crucial in the angiogenesis process, which is essential for tumor progression. An increased level of expression and higher activity of MMP-2 is observed with increased vascularization of the metastases of lung cancer in the central nervous system [85]. It has been shown that MMP-2 may affect tumor viability and invasiveness also by regulating lymphangiogenesis [80]. In contrast, MMP-9 and other metalloproteinases play a role in angiogenesis by promoting mobilization of VEGF (vascular endothelial growth factor) [80]. Berberine inhibits metastases by impeding angiogenesis through the effects on MMP-2 and MMP-9. However, inhibition of angiogenesis by berberine does not only affect metalloproteinases.

In B16F-10 melanoma cells exposed to berberine, decreased expression of genes encoding angiogenesis-promoting factors, i.e., COX-2, HIF (hypoxia induced factor) and VEGF were reported [58]. Additionally, in the liver cancer HepG2 cell line and the gastric adenocarcinoma SC-M1 cell line, berberine was found to inhibit cell proliferation, migration, vascular endothelium formation and VEGF expression [86]. In breast cancer cells, berberine reduced the expression of VEGF and fibronectin by inhibiting the PI-3K/AKT pathway [84]. Inhibition of NF-κB in tumor cells is also one of the mechanisms of a berberine-induced decrease in the expression of VEGF and IL-8 [29].

Prevention of metastasis by inhibition of angiogenesis by berberine was confirmed in vivo [87]. In tests on mice, berberine reduced tumor vasculature by inhibiting the activity of factors responsible for angiogenesis, e.g., VEGF, inflammatory mediators: IL-6, IL-1β, TNF-α and a factor stimulating the formation of macrophage colonies—GM-CSF (granulocyte macrophage colony-stimulating factor). Berberine was also reported to inhibit the activity of transcription factors responsible for angiogenesis, i.e., NFκB, c-Fos, CREB (cAMP response element-binding protein) and ATF-2 (activating transcription factor 2) [54,82].

### 4.7. Inhibition of Epithelial-To-Mesenchymal Transition

Another facet of the antimetastatic action of berberine is its effect on E- and N-cadherin. Berberine influences the expression of the E-cadherin and N-cadherin proteins and the effect is time and dose dependent. E-cadherin and N-cadherin are closely related to the migration and invasion of cancer cells. E-kadherin is responsible for the structural integrity of epithelial cells [88]. It is a marker of TGF-β1-induced epithelial-to-mesenchymal transition. This process leads to increased cell motility. TGF-β (transforming growth factor beta 1) is a cytokine mediating progression by enhancing the epithelial-to-mesenchymal transition process. Berberine has been proven to inhibit the TGF-β1-induced epithelial-to-mesenchymal transition process and an elevated level of E-cadherin is a marker of this process [88]. MMP-2 activates TGF-β [89]; hence, the influence of berberine on the process of epithelial-to-mesenchymal transition is probably based on the above-described reduction of the level of metalloproteinase 2 growth factors [90] and the release of growth factors from outside the extracellular matrix, such as TGF-β [91,92].

By affecting metalloproteinases, berberine may indirectly influence the occurrence of apoptosis. For example, in human melanoma cells with expressed integrin αvβ3, the degradation of type I of collagen by MMP-2 may reveal a binding site with integrin αvβ3 in these cells. Signaling by this integrin is essential for the viability of melanoma cells and growth in the collagen matrix, thus it potentially protects melanoma cells from apoptosis [81].

MMP inhibitors possess high potential for improving cancer treatment by slowing the process of cancer invasion [79]. Phase I of clinical trials has shown that MMP inhibitors produce minimal adverse side effects. This way, berberine used as a MMP inhibitor seems to be a potential anticancer agent.

### 4.8. Anti-Inflammatory Activity

Anti-inflammatory activity is also one of the important effects rendering berberine a promising anticancer and protective agent. Berberine has anti-inflammatory activity in vitro and in vivo and inhibits the transcription of genes such as IL-1, TNF- and IL-6, decreasing the level of inflammatory proteins. Berberine suppresses the expression of cyclooxygenase 2 and prostaglandin E2 [29]. Studies have also demonstrated that berberine prevents production of IL-8 in tumor cells and blocks the NF-κb signaling pathway [58]. Studies have shown that berberine also inhibits the elevation of NO and TNF-α [29,82]. In colorectal cancer cells, berberine also inhibits COX-2 transcriptional activity, which is significantly increased in this type of cancer [78].

### 4.9. β-Catenin Expression

Berberine acts on β-catenin, whose mutations and overexpression are associated with many cancers, including colorectal carcinoma, breast tumors, ovarian and endometrial cancer. In colon cancer cells, the expression of its mRNA, is downregulated by berberine. Berberine efficiently inhibits the nuclear level of β-catenin by increasing adenomatous polyposis coli protein and β-catenin interactions [37]. It stimulates the expression of adenomatous polyposis coli protein and regulates β-catenin negatively [29,54].

### 4.10. Inhibition of Carcinogenesis Combined with Metabolism of Lipids

The influence on the metabolism of fats and lipids is one of the mechanisms of berberine action in the aforementioned metabolic diseases, but it also seems to be very important in malignancies, especially those of the digestive system. It has been described that berberine can act by inducing apoptosis via reduction of FABP expression and accumulation of fatty acids in gastric cancer [93] and by downregulation of key lipogenic enzymes in colon cancer. Berberine targets the SREBP-1 cleavage-activating protein-1/sterol receptor element-binding protein-1 (SCAP/SREBP-1) pathway driving lipogenesis, inhibits the pathway and thus results in downregulation of lipogenic enzymes. Downregulation of key lipogenic enzymes, leading to suppression of lipid synthesis, which is linked to cell proliferation through the Wnt/β-catenin pathway, has been described as one of the anticancer mechanisms of berberine [53]. Similarly, the influence on JNK kinases described above is important in anticancer and chemopreventive activity based on the influence on lipid metabolism [94,95].

Anticancer activity of berberine has beed summarised in Table 1.

## 5. Supportive Action of Berberine—Sensitization and Drug Resistance

The promising applications of berberine are limited by the poor pharmacokinetics of the compound. Nevertheless, berberine seems to be very effective and possible to use in combined treatment with other chemotherapeutics or therapies. In terms of sensitization, berberine can be considered as a photosensitizing agent in photodynamic therapy. In a study on renal cancer with human renal tubular epithelial cells derived from the normal kidney HK-2 cell line and human cell lines ACHN and the 786-O cell line, berberine increased the level of autophagy and the level of reactive oxygen species. Berberine induced apoptosis in these cells by induction of caspase 3. Low expression of genes of human telomerase reverse transcriptase and vascular endothelial growth factor-D was observed after combined exposure. Furthermore, polo-like kinase 3 exhibited overexpression after treatment with berberine combined with photodynamic therapy [96]. It has also been described that berberine is a promising potential sensitizer for the radiotherapy of hepatocellular carcinoma, where Nrf2, i.e., a master transcription factor in oxidative damage, is required for this effect of berberine [97,98].

Sensitization of breast cancer cells to chemotherapeutics seems to be the most promising approach as regards the sensitizing activity of berberine, but other cells, e.g., hepatocellular carcinoma, leukemic, ovarian and lung cancer cells were also described. In the breast cancer MDA-MB-231 cell line, berberine sensitized cancer cells to methyl methanesulfonate, cisplatin and camptothecin. Simultaneously, no synergistic effects with hydroxurea and olaparib were observed. The mechanism of sensitization is probably based on interference of berberine with DNA repair protein—XRCC1 (X-ray repair cross-complementing protein 1)—mediated base excision. As suggested by Gao et al., this may be a basic mechanism in the use of berberine in the therapy of breast cancer [99].

Combination therapy of berberine and cisplatin markedly enhanced the death of other ovarian cancer cells by inducing necroptosis and apoptosis in the ovarian cancer OVCAR3 cell line and primary ovarian cancer cells with a dose- and time-dependent effect. The apoptosis was caspase-dependent, while the necroptosis accompanied the activation of the RIPK3-MLKL pathway [100]. In combination with gefitinib in non-small-cell lung cancer, berberine inhibits epithelial–mesenchymal transition [101].

The combined treatment with doxorubicine showed in vitro and in vivo reduction of repopulation in hepatocellular carcinoma treatment, achieving a synergic effect inhibiting the Caspase-3-iPLA2-COX-2 pathway [102] and a better therapeutic p53-independent effect than doxorubicin alone in leukemic cells [103]. Berberine was found to reverse doxorubicin resistance in MCF-7 and (ADR)-resistant MCF-7 breast cancer cells by inhibiting autophagy, which makes it a promising agent for clinical application in breast cancer treatment. As a suppressor of autophagy, berberine inhibits formation of autophagosome in MCF-7/ADR cells by blocking the accumulation of protein LC3II associated with autophagy. This results in reversal of doxorubicin resistance and reduced cell proliferation. Cellular accumulation of p62 and inhibition of autophagy were also observed. Berberine acted by modulating the PTEN/Akt/mTOR signaling pathway [104].

The supportive activity of berberine in combined treatment is not clarified and needs further research. Importantly, the formulation seems to play a key role. Doxorubicin conjugated to poly (lactic-co-glycolic acid) and used for encapsulation of berberine induced cell cycle arrest in the sub-G1 phase, significant depolarization of the mitochondrial membrane and necrosis in the breast cancer MDA-MB-231 cell line. In vivo studies revealed a very high increase in the half-life and in the plasma drug concentration in such a mode of distribution [105]. Additionally, a nanocarrier hyaluronic acid-conjugated Janus formulation codelivering doxorubicin and berberine exhibited enhanced tumor accumulation and biocompatibility [102]. Maiti et al. described that berberine in combination with curcumin as solid lipid curcumin particles had higher bioavailability and showed higher anticancer effects in cultured cancer cells than in the natural state. In comparison of single and combined treatment of solid lipid curcumin particles and berberine in the human neuroblastoma SH-SY5Y cell line and the human glioblastoma U-87MG and U-251MG cell line, a higher rate of glioblastoma cell death, enhanced fragmentation of DNA, substantially decreased levels of ATP and reduced potential of mitochondrial membrane were observed in the cotreatment of solid lipid curcumin particles and berberine [106].

Novel formulations seem to be crucial in the sensitizing efficacy described above and are important in increasing berberine efficacy itself. For example, the novel berberine complex targeting telomerase appeared to induce dysfunction of mitochondria, damage of the DNA telomere and cell-cycle arrest [107]. Berberine preloaded into folic acid, targeting Janus gold mesoporous silica nanocarriers, exerted a highly potent antitumor effect in patients with liver cancer and ensured good biosafety and effective protection of normal tissues [108]. It is worth mentioning that berberine is a compound with proven ability to bind to G-quadruplexes. Currently, berberine has also been proven to increase the affinity of iminopyrenyl-β-cyclodextrin for the DNA duplex, which may be important in terms of developing new therapeutic formulations [109].

## 6. Preventive Action of Berberine

The preventive potential of berberine seems to be very promising due to the action towards cancer stem cells and the anti-inflammatory properties of the compound. In combination with d-tri-phenyl-phosphonium at concentrations that are toxic only to cancer cells, berberine effectively decreased transmission of cancer stem cells, which can be important in prevention of many malignancies, as these cells are involved in initiation of tumor and metastatic dissemination and play an important role in cancer therapy resistance [110].

As described in research on colon cancer, berberine can be considered as a preventive agent due to its supportive action in colitis leading to colon cancer in 5% of patients with this disease. Berberine markedly decreased the Geboes grade in colitis in vivo. On the other hand, it had an insignificant effect on other tissues or blood markers related to inflammation and cell growth, while the combination of mesalamine and berberine enhanced the anti-inflammatory effects of mesalamine on colonic tissue in patients with ulcerative colitis [111]. On the other hand, other authors suggest that the anti-inflammatory activities of berberine itself make it an opportunity to prevent cancers associated with inflammation such as colorectal cancer [112].

## 7. Challenge to New Derivatives and Formulations of Berberine

Although chloride or sulphate salts of berberine are better soluble and are therefore used clinically [3], the low bioavailability and poor pharmacokinetic parameters of berberine are still the main challenge in its the potential usage. Next to novel formulations, development of new derivatives with similar biological activity based on the same mechanisms but not limited by low pharmacological parameters seems to be the most important target. Compounds based on berberine that will be effective at lower concentrations and have stronger biological activity are currently being intensively investigated. Especially the latest papers from 2019 and 2020 provide information on derivatives, giving a promising view, as a significant relationship between structure and activity was observed [113]. New derivatives exhibit similar parameters of biological activity and are promising for further research [114].

To date, berberine-12-amine derivatives have been evaluated in terms of the inhibition of the growth of human cancer cell lines. Quaternary 12-aminoberberine chloride, quaternary 12-nitroberberine chloride, tertiary 12-aminotetrahydroberberine and 12-aminoberberine derivatives with different states of reduction showed growth inhibition activities in human cancer cells: colorectal cell line HCT-8, gastric cancer cell line BGC-823, liver cancer cell line Bel7402, cervical cancer cell line HeLa and lung cancer cell line A549. Quaternary berberine-12-N,N-di-n-alkylamine chlorides are significantly stronger than reduced counterparts. The activities increased with the elongation of the n-alkyl carbon chain of 12-N,N-di-n-alkylamino in the range of about 6–8 carbon atoms. The activities decreased with the elongation of the n-alkyl carbon chain when the carbon atom numbers exceeded 6–8. The activities of the tertiary amine structure were significantly higher than that of the secondary amine structure [113].

13-alkyl-substituted derivatives of berberine are more active than berberine itself towards human cancer cell lines. In radiotherapy-resistant MDA-MB-231 cells, there were lower levels of proapoptotic genes and higher levels of antiapoptotic genes than in the breast cancer MDA-MB-231 cell line treated with 13-ethylberberine. In both cell lines, reduced proliferation and colony-formation was described after exposure to this derivative. 13-ethylberberine induced apoptosis by promoting mitochondrial and intracellular reactive oxygen species and by regulating proteins involved in the intrinsic pathway of apoptosis but not in the extrinsic apoptosis pathway. Such derivatives occurred to be similarly effective and seem to be candidates in therapeutic strategies characteristic for berberine but exact comparison to berberine in terms of bioavailability and distribution are still needed [115].

The synthesis and analysis of 9-O-substituted derivatives of berberine also gave promising effects. Berberine derivative (9-(3-bromopropoxy)-10-methoxy-5,6-dihydro-[1,3]dioxolo[4,5-g]isoquino[3,2-a] isoquinolin-7-ylium bromide) was found to have 30-fold higher antiproliferative activity and 6-fold higher apoptosis-inducing activity in leukemia cells, compared to berberine [116].

Additionally, the assessment of berberine derivatives with cis-substituents at positions C9 and C13 [117,118] or 13-[CH2CO-Cys-(Bzl)-OBzl]-berberine [119] and a series of berberine derivatives with modified position 9-O [120,121] was described. The methylene-dioxy and methoxyl groups in berberine seem to be especially important for the anticancer activity exhibited by its derivatives [122].

The absorption of bioactive berberine (Quillaja extract emulsified berberine) in humans is in the recruiting phase of a currently running clinical trial [123].

## 8. Berberine in Clinical Trials

Due to the wide range of effects and safety, berberine has been assessed clinically. There are many ongoing trials (Table 2) and even more interesting ones are going to start soon. Berberine was clinically assessed in schizophrenia and its metabolic and cardiovascular effects were assessed in the metabolic syndrome, obesity, diabetes mellitus type 2 and insulin-resistance. The compound was clinically assessed in non-alcoholic fatty liver disease, dyslipidemia and hypercholesterolemia. It was also clinically assessed in cases of colorectal adenoma, gastric ulcer, chronic gastritis and gastric cancer [54]. Its anticancer activity was studied to assess mitigation of the effects of radiation therapy in the treatment in patients with lymphoma and cervical cancer. In patients with non-small cell lung cancer during radiation therapy, berberine was found to protect lung cells from ionized radiation-induced damage [42]. In patients with glioma, it selectively sensitized tumor cells to ionizing radiation, while healthy cells remained at the same level of sensitivity [87]. There are many completed clinical trials assessing berberine. The main target is its activity lowering cholesterol and glucose levels. The nutraceutical potential of berberine combined with some natural compounds seems to be very interesting.

In patients with type 2 diabetes mellitus berberine significantly improved the level of fasting blood glucose, the level of postprandial blood glucose and the level of glycosylated hemoglobin and decreased insulin resistance. It has been estimated that berberine exerts a beneficial effect on blood glucose control comparable to that of metformin. Berberine appears to have an advantage over rosiglitazone in improving the level of fasting blood glucose. The combination therapy of berberine with oral hypoglycemic drugs may arouse high hopes. However, the efficacy of berberine should be further evaluated in a larger patient population of patients with type 2 diabetes mellitus [124].

As a nutraceutical, berberine with chromium picolinate, inositol, curcumin and banaba was clinically assessed in patients with fasting dysglycemia. It was reported to be helpful in improving glyco-metabolic compensation and total cholesterol and triglyceride value and in reducing inflammatory status in patients with dysglycemia. Reduced fasting and post-prandial level of glucose in plasma was observed after administration of the nutraceuticals. There was also a decrease in the level of fasting plasma insulin and in the level of glycated hemoglobin. Moreover, in patients with fasting dysglycemia after 3 months of application of combined nutraceutical therapy, the level of high-sensitivity C-reactive protein was reduced [125]. Furthermore, berberine was clinically assessed in metabolic syndrome. In metabolic syndrome in schizophrenia, the effect of berberine administration on insulin secretion, insulin sensitivity and metabolic syndrome was evaluated. A significant decrease in waist circumference and remission of the presence of metabolic syndrome was noted in patients after the administration of berberine. A marked decrease in systolic blood pressure, triglycerides, area under the curve of glucose, area under the curve of insulin and insulinogenic index and an increase in the Matsuda index were reported. According to the clinical trials, administration of berberine abolishes the metabolic syndrome and decreases waist circumference. It also decreases insulin secretion and the levels of triglycerides Simultaneously, it increases insulin sensitivity in patients with metabolic syndrome in schizophrenia [126]. Additionally, the hypoglycemic effect of berberine and bifidobacterial administration in patients with prediabetes or diabetes mellitus was assessed and it was proved that berberine and bifidobacteria may be supportive in the treatment of diabetes [127].

An important and clinically assessed aspect of berberine activity is its cholesterol lowering effect. The efficacy and safety of a nutraceutical combination consisting mainly of red yeast rice extract, berberine and policosanols were assessed in patients with low to moderate risk of hypercholesterolemia. This combination associated with a hypolipidemic diet reduced total cholesterol and LDL-C levels [128]. Assessed in non-alcoholic fatty liver disease, comparison of berberine supplementation to lifestyle intervention and berberine treatment plus lifestyle intervention resulted in a strong reduction of hepatic fat content accompanied by improvement in serum lipid profiles and body weight. Berberine reduced body weight and the lipid profile more effectively than pioglitazone. The adverse events were mild and mainly affected the digestive system. Berberine was clinically proved to improve parameters in patients with non-alcoholic fatty liver disease and related metabolic disorders. Its therapeutic effect may involve regulation of hepatic lipid metabolism [129]. The compound has also been evaluated clinically in menopausal women at risk of dyslipidemia. In combination with compounds of plant origin such as chlorogenic acid and tocotrienols, a reduction in LDL and total cholesterol levels was observed after three months of supplementation; however, the influence of berberine on menopausal symptoms requires further research [130].

Berberine-containing quadruple therapy was assessed as an alternative treatment for *Helicobacter pylori* eradication was assessed in comparison to bismuth-based quadruple regimen, which is an alternative for *Helicobacter pylori* eradication due to the increasing antimicrobial resistance. The eradication rates in berberine groups were at a similar level as in the bismuth group. There was also no statistically significant difference in the incidence of adverse events. Therefore, berberine can be an alternative for *Helicobacter pylori* eradication [131].

The chemopreventive potential of colorectal adenoma prevention and recurrence was assessed clinically. Patients with colorectal adenomas subjected to complete polypectomy received berberine twice daily. The treatment proved to be safe and effective. Reduced risk of recurrence of colorectal adenoma was found, showing berberine as an option for chemoprevention after polypectomy [127].

Currently, berberine is at the stage of clinical trials with its cardiovascular and metabolic effects. It is clinically compared with metformin as an agent in therapy for metabolic syndrome in patients with schizophrenia and clinically assessed as an adjuvant treatment for patients with schizophrenia and its spectrum, patients with other psychotic disorders or metabolic syndrome X. Berberine is in clinical trials assessing its suitability for hyperglycemic clamp, in diabetes mellitus and as a preventive agent in induced nephropathy in patients with Type 2 diabetes mellitus accompanied by chronic kidney disease. The beneficial berberine effects on the risk factors of cardiovascular diseases and its antiplatelet effect are clinically assessed in patients after elective percutaneous coronary intervention in coronary artery disease. In oncological patients, berberine is currently being assessed as a preventive agent in patients with colorectal adenomas with previous colorectal cancer treated with gefitinib. In the nearest future, berberine will be clinically studied in patients with lung adenocarcinoma, corticosteroid-resistant or relapsed immune thrombocytopenic purpura and patients with graft pain and gingival recession. Trials with diabetic patients will be continued as well [123].

## 9. Summary

Berberine is one of the most interesting and promising natural agents currently due to its proven biological activity, especially in biochemical pathways important in apoptosis, carcinogenesis and metastasis. As a natural compound with low toxicity towards healthy cells, berberine shows great potency in the treatment of many clinical stages, e.g., metabolic disorders and related symptoms, inflammation and its after-effects, or cancer prevention and combined cancer treatment. However, there are limitations in the dissolution, absorption and biodistribution of berberine. Currently, development of new formulations and new derivatives is a very important trend in berberine research to overcome its limitations in clinical application [34,132].

## Figures and Tables

**Figure 1 toxins-12-00713-f001:**
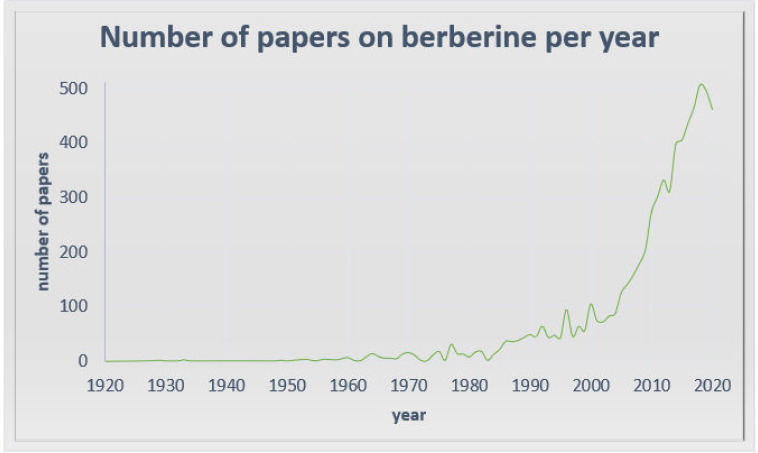
Structure of the number of publications on berberine based on the PubMed database: https://pubmed.ncbi.nlm.nih.gov.

**Table 1 toxins-12-00713-t001:** Molecular mechanisms of anticancer activity of berberine.

Activity	Molecular Targets	References
Cell cycle arrest		
	Induced G0/G1 phase arrest	[47,51,52,53]
	↓cyclin B1 expression	
	Induced G1 phase arrest	
	↑geneBTG2 expression	
	↑p53 protein expression	[47,48,49]
	↓cyclin B1 expression	
	↑Weel1 expression	[47,54]
	↓cyclin D1 expression	[47]
	Induced G2 phase arrest	
	↓cyclin B1 expression	
	↑Weel1 expression	[47,54]
	Induced G2/M phase arrest	
	REV3-gene dependent	[55,56]
Apoptosis induction		
	genes	
	↑CASP3 expression	
	↑CASP8 expression	
	↑CASP9 expression	
	↑BAX expression	
	↑BAK1 expression	
	↑BIK expression	
	↓BNIP3 expression	
	↓BNIP1 expression	
	↓BCL2 expression	
	↓BCL2L2 expression	[43]
	proteins	
	↑caspase-3 expression	
	↑caspase-9 expression	
	↑Bax expression	[33,45]
	↓Bcl-2 expression	[33]
	↑p-53 expression	
	↑ Rb expression	
	↑ATM expression	
	Caspase-8 expression	
	↑BID expression	
	↑TNF expression	
	↓c-IAP1 expression	
	↓XIAP expression	
	↓Bcl-X expression	
	↓Survivin expression	[22,58,59]
	TNF-alpha receptors	
	↑DR4 activation	
	↑DR5 activation	[54,60]
Influence on MAPK		
cell type depending	↑MAPK activity	[20,21,61]
	↑miR-19a/TF/MAPK pathway activity	
	↑miR-19a level	
	↓TF level	[21]
	↑ERK1/2 phosphorylation	[62]
	↓ERK1/2 phosphorylation	[19]
	↓p38 MAPK phosphorylation	[19,62]
	↑JNK phosphorylation	[62]
	↓JNK phosphorylation	[19]
	↓JNK/p38 MAPK pathway activity	[22]
	↑JNK/p38 MAPK pathway activity	[20]
	↑cell cycle inhibitor p21 (CIP1/WAF1) induction	
	↑p53 level	[20]
	↑FOXO3a induction	
Transcription Regulation		
	↑/↓AP-1 protein activity (cell type depending)	[39,43,78]
	↓ C-fos proto-oncogene expression	[54]
Inhibition of metastasis		
	proteins	
	↓MMP-2 level	
	↓MMP-9 level	[54,82,83]
	↓p-STAT3 phosphorylation	
	↓p-STAT3 transfer to nucleus	
	↓p-JAK2 phosphorylation	
	Interruption of COX2/JAK/STAT signaling pathway	[23,29]
Inhibition of angiogenesis		
	genes	
	↓VEGF expression	
	↓COX-2 expression	
	↓HIF expression	[58]
	proteins	
	↓VEGR expression and activity	
	↓IL-1β activity	
	↓IL-6 activity	
	↓TNF-α activity	
	↓GM-CSF activity	
	↓NF-κB activity	
	↓c-Fos activity	
	↓CREB activity	[54,82]
	↓PI-3K/AKT pathway	[84]
Inhibition of epithelial-to-mesenchymal transition		
	↓TGF-β1 release	
	↑E-cadherin level	[90,91,92]
Anti-inflammatory activity		
	genes	
	↓IL-1 transcription	[29]
	↓TNF transcription	
	↓IL-6 transcription	
	proteins	
	↓COX2 expression	[29]
	↓COX2 transcriptional activity	[78]
	↓IL-8 tumor cells production	[58]
	↓NF-κB signaling pathway	[22,58]
	↓NO elevation	
	↓TNF-α elevation	[29,82]
Inhibition of carcinogenesis combined with metabolism of lipids		
	↓levels of fatty acids in cancer cells	[93]
	↓SCAP/SREBP-1 pathway	
	↓activity of lipogenic enzymes	
	↓JNK phosphorylation	
	↓lipid synthesis	[53]

↓decreasing; ↑increasing.

**Table 2 toxins-12-00713-t002:** Currently running and active clinical trials based on: https://clinicaltrials.gov.

Title	Status	Conditions	Interventions	Phase	Measures	Enrollment	Age (Years)	Completion:
**A Research of Berberine Hydrochloride to Prevent Colorectal Adenomas in Patients with Previous Colorectal Cancer**	Recruiting	Colorectal Adenomas	Berberine hydrochloride and Placebo	Phase 2 Phase 3	Cumulative colorectal adenoma incidence rate during Berberine hydrochloride or placebo treatment in patients with a history of colorectal cancer Cumulative numbers or diameters of new colorectal adenomas during Berberine hydrochloride or placebo treatment in patients with a history of colorectal cancer	1000	18–80	March 2021
**Comparison of Berberine and Metformin for the Treatment for MS in Schizophrenia Patients**	Recruiting	Schizophrenia Metabolic Syndrome	Berberine Metformin	Phase 4	Fasting blood samples for Fasting blood glucose Triglyceride High-Density Lipoprotein Waist circumference Blood pressure including systolic and diastolic pressure Body mass index Total Cholesterol C reactive protein Interleukin-1,Interleukin-6,tumor necrosis factor(TNF)	100	18–65	December 2019
**Primary Chemoprevention of Familial Adenomatous Polyposis with Berberine Hydrochloride**	Recruiting	Colorectal Adenomas	Berberine hydrochloride Placebo	Phase 2 Phase 3	Cumulative numbers and diameters of colorectal adenomas during Berberine hydrochloride or placebo treatment in patients with familial adenomatous polyposis	100	18–65	December 2020
**Effect of Berberine Versus Metformin on Glycemic Control, Insulin Sensitivity and Insulin Secretion in Prediabetes**	Active, not recruiting	Prediabetes Impaired Fasting Glucose Impaired Glucose Tolerance	Berberine Metformin	Phase 4	Fasting glucose levels postprandial glucose levels Glycosylated hemoglobin Total insulin secretion First phase of insulin secretion Insulin sensitivity Body Weight Body Mass Index Body fat percentage Waist circumference and 8 more	28	31–60	August 2020
**Effect of Berberine for Endothelial Function and Intestinal Microflora in Patients with Coronary Artery Disease**	Active, not recruiting	Stable Coronary Artery Disease Percutaneous Coronary Intervention	Berberine Aspirin Clopidogrel Statin	Phase 1 Phase 2	Endothelial function measured by flow mediated dilation Gut microbiome Fecal metabolomics profile measurement Blood lipid levels Inflammatory factor levels Blood glucose levels	24	18–75	December 2020
**A Mechanistic Randomized Controlled Trial on the Cardiovascular Effect of Berberine**	Recruiting	Cardiovascular Risk Factor	Berberine Placebo	Phase 2 Phase 3	lipid profile blood pressure thromboxane A2 testosterone body mass index waist hip ratio fasting glucose fasting insulin liver function	84	20–65	June 2020
**Combination of Danazole With Berberine in the Treatment of ITP**	Active, not recruiting	Corticosteroid-resistant or Relapsed ITP	Berberine plus danazol	Phase 2	The Count of Participants That Achieved 6-month Sustained Response the Count of Participants That Had Adverse Events the Count of Participants That Achieved Initial Response	55	18–80	June 2021
**Antiplatelet Effect of Berberine in Patients After Percutaneous Coronary Intervention**	Recruiting	Coronary Artery Disease Percutaneous Coronary Intervention	Berberine Standard treatment Aspirin Clopidogrel	Phase 4	P2Y12 reaction unit Platelet reactivity index Urinary 11-dehydrothromboxane B2 (11-dHTXB2)	64	18–70	December 2020
**Berberine Chloride in Preventing Colorectal Cancer in Patients with Ulcerative Colitis in Remission**	Active, not recruiting	Ulcerative Colitis	Berberine Chloride Placebo	Phase 1	Incidence of organ toxicity Clinical efficacy of berberine chloride measured using the UCDAI score plasma markers of inflammation colorectal tissue biomarkers expression gene methylation status blood berberine chloride concentration Severity of histologic inflammation	18	18–70	December 2020
**Berberine Prevent Contrast-induced Nephropathy in Patients With Diabetes**	Recruiting	Diabetes Mellitus Chronic Kidney Disease	Berberine	Phase 4	Contrast-induced nephropathy Major adverse renal events	800	18 and older	December 2020
**Berberine as Adjuvant Treatment for Schizophrenia Patients**	Recruiting	Schizophrenia Spectrum and Other Psychotic Disorders Metabolic Syndrome x	Berberine Placebos Antipsychotic Agents	Phase 2 Phase 3	Weight gain Changes in body mass index Changes in waist circumference Changes in blood pressure Changes in triglycerides Changes in total cholesterol Changes in high-density lipoprotein Changes in low-density lipoprotein Changes in fasting glucose Changes in insulin	120	18–65	May 2021
**Evaluating the Tolerability and Effects of Berberine on Major Metabolic Biomarkers: A Pilot Study**	Recruiting	Metabolic Syndrome	Berberine Identical Placebo	Not Applicable	LDL Cholesterol Hemoglobin A1c Number of participants with adverse events	40	18 and older	December 2021
**Efficacy and Safety of Berberine in Non-alcoholic Steatohepatitis**	Recruiting	Non-alcoholic Steatohepatitis	Behavioral: Lifestyle Placebo	Phase 4	Improvement in histologic features of nonalcoholic steatohepatitis by NAFLD activity score Improvement in the composites of NAFLD activity scores for steatosis, lobular inflammation, hepatocellular ballooning Improvement in liver histological fibrosis staging Resolution of NASH anthropometric measures blood biochemistry liver fat content serum cytokeratin 18 (CK-18) in U/L	120	18–75	July 2021
**Study on the Efficacy and Gut Microbiota of Berberine and Probiotics in Patients with Newly Diagnosed Type 2 Diabetes**	Active, not recruiting	Type 2 Diabetes	Berberine hydrochloride ProMetS probiotics powder	Phase 3	HbA1c Gut microbiome Fasting glucose levels 2-h postprandial glucose levels Fasting insulin levels 2-h postprandial insulin levels Serum Triglycerides Serum total Cholesterol Serum HDL-c Serum LDL-c	400	20–69	May 2019
**Effect of Mebo Dressing Versus Standard Care on Managing Donor and Recipient Sites of Split-thickness Skin Graft**	Recruiting	Burns	Moist Exposed Burn Ointment (MEBO-sesame oil, beta-sitosterol, berberine and other small quantities of plant ingredients)	Phase 1	Wound Healing Assessment Recovery Time Rate of Infections Pain Assessment Total Treatment Costs Rate of Complications Quality of Life	40	2–60	July 2019

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
