# Peer review of "Biological Activity of Berberine—A Summary Update"

_toxins, 2020, doi:10.3390/toxins12110713_

Round 1
Reviewer 1 Report
The manuscrpt is now much well organized and readable. The inclusion of Tables permits to identify the area of interest.
Author Response
Dear Reviewer,
Thank you very much for reading my work. Thanks to the previous comments and hard work, I was able to receive a positive review for which I am very grateful. I hope this work will be of use to those seeking knowledge about berberine. I wish you much all the best.
Kind Regards
Reviewer 2 Report
The authors have taken into account my previous criticisms. I recommend to accept the resubmitted version of the manuscript.
Author Response
Dear Reviewer,
Thank you very much for reading my work. Thanks to the previous comments and hard work, I was able to receive a positive review for which I am very grateful. I hope this work will be of use to those seeking knowledge about berberine. I wish you all the best.
Kind Regards
Reviewer 3 Report
Opinion on the revised version of the manuscript entitled "Biological Activity of Berberine –a Summary Update".
Authors of the above-mentioned paper substantially improved the manuscript. Is seems suitable for acceptance and publication in Toxins.
Minor point: the content of Fig 1 (Structure of the number of publications on berberine) seems to different from from Fig 1 in the oridinal version. The maximum value is higher in the newer version. What is the reason? The source of the data should be indicated.
Author Response
Dear Reviewer,
Thank you very much for reading my work.
As regards the content of Fig 1 (Structure of the number of publications on berberine) it has been corrected. The maximum value is now 506 what concerns number of publications in 2018 according to the PubMed data base.
Thanks to your previous comments and hard work, I was able to receive a positive review for which I am very grateful.I hope this work will be of use to those seeking knowledge about berberine. I wish you much all the best.
Kind Regards
This manuscript is a resubmission of an earlier submission. The following is a list of the peer review reports and author responses from that submission.
Round 1
Reviewer 1 Report
This paper is of potential general interest but the organization of the manuscript should likely be improved to facilitate the reader. The Authors mention several times clinical trials but the review instead is discussing maily in vitro data. It is suggested that in vitro from in vivo data should be separate. Furthermore, the clinical trials should have a special section and attention.
The organization of the paper should consider using tables to reduce the spread of information that frequently become confusing for the reader.
Reviewer 2 Report
The review entitled The Biological Activity of Berberine – The Summary Update provides historical overview and highlights recent progress made to understand the molecular mechanisms of berberine action. The review is informative and understandable.
Although the manuscript is suitable for this journal, it has some shortcomings as follows:
The manuscript needs a round of editing by a native English speaker.
References need to be checked.
Lanes 6-7, 38-39 and 414: The sentences should be revised. Berberine is actually toxic also to normal cells. Therapeutic window of berberine in most cases is narrow and depends on type of cells which are treated.
The overview scheme and table have to be included to provide a summary of anticancer mechanisms of berberine. This will help for a wide range of readership.
Reviewer 3 Report
Opinion on the manuscript entitled „ The Biological Activity of Berberine – The Summary Update” submitted to Toxins for publication
Manuscript ID: toxins-946307
The manuscript is supposed to be a comprehensive review on the pharmacology and potential therapeutic use of the alkaloid berberine. The manuscript contains substantial amount of published results, extracted from 85 articles. However, there are some basic arguments against its publication. These are the followings:
Major points:
- The most relevant objection against the publication of the manuscript in its current form is the way of interpretation of the presented results. The Authors simply listed the main findings recently published about the pharmacological profile of berberine. No mechanistic explanations are presented, no possible mechanisms of action are described. There is no novelty synthesized from previous results. There is a recent review on the pharmacology of berberine published by Liu et al: A Natural Isoquinoline Alkaloid With Antitumor Activity: Studies of the Biological Activities of Berberine. Front Pharmacol (2019) doi: 10.3389/fphar.2019.00009. Though this 2019 paper is restricted to anticancer properties of the alkaloid but gives more mechanistic insight.
- There are some issues concerning the originality of the paper. Lines 203-206:
“E-cadherin and N-Cadherin proteins are closely related to cell migration and invasion. Moreover, E-cadherin is not only an important mediator that regulates cell-celladhesion, but also an important molecule in the maintenance of the morphology and structural integrity of epithelial cells”.
Pages 2-3 of the above-mentioned Front Pharmacol (2019) article:
“E-cadherin and N-Cadherin proteins are closely related to cell migration and invasion. Moreover, E-cadherin is not only an important mediator that regulates cell-cell adhesion, but also an important molecule in the maintenance of the morphology and structural integrity of epithelial cells”
This kind of word by word identity is not considered acceptable. - There are some discrepancies between statements in the text and the cited references. E.g. the paragraph lines 196-201 details the action of berberine in lung cancer but the inserted citation (29) is something about liver cell line: Fukuda et al: Inhibition of activator protein 1 activity by berberine in human hepatoma cells. Planta Med (1999) doi: 10.1055/s-2006-960795.
Minor points:
- There are many typing errors in the text. Some examples: “Berberineis” (line 17), “activityiscurrently” (line 18), “concentrationandtime-dependentmanner” (line 202).
- The definition of the cited references is not consequent. Some of the journals are abbreviated (e.g. 7, 9, 17, 60) for many others the full names are given (e.g. 3, 8, 59).
- Though the writer of the present review is not a native English speaker and therefore not competent in the evaluation of style of the text but he has the impression that a careful grammar and stylistic check, presumable by a native English speaker, would substantially improve the overall values of the manuscript.
- Figure 1: the direction of the time axe is unusual. The usual presentation, in which the past is on the left side, would be more convenient for the readers.
Based on these objections the rejection of the manuscript is suggested.